# Challenges and Choices in Breastfeeding Healthy, Sick and Preterm Babies: Review

**DOI:** 10.3390/healthcare12232418

**Published:** 2024-12-02

**Authors:** Susanne H. Bauer, Harald Abele, Joachim Graf

**Affiliations:** 1Institute for Health Sciences, Department of Midwifery Science, University Hospital Tuebingen, 72076 Tuebingen, Germany; shbauer@web.de (S.H.B.); harald.abele@med.uni-tuebingen.de (H.A.); 2Department of Women’s Health, University Hospital Tuebingen, 72076 Tuebingen, Germany

**Keywords:** breastfeeding, challenges, choices, sick children, preterms

## Abstract

Although breastfeeding is associated with many health-related benefits for both mothers and children, the WHO recommendation for exclusive breastfeeding is not achieved by the majority in any WHO region. This paper aims to present the current state of research on challenges and choices in breastfeeding healthy, sick and preterm babies. The research was organized as a systematic search in PubMed and the study was performed as a narrative review after applying the PRISMA protocol. Finally, *n* = 57 studies were included. Both barriers and support factors emerge as a complex interaction of individual, group and societal factors, the precise understanding of which is relevant to increasing breastfeeding rates in the future. Knowledge as well as practical skills proved to be generally helpful, whereas the lack of breastfeeding support for mothers, who are often separated from their premature babies in hospital, was identified as a key risk factor for this subgroup. Appropriate training for healthcare professionals can improve the situation as a result. After discharge, workplace-related barriers are of major concern to allow further breastfeeding when maternity leave ends. Thus, the promotion of breastfeeding must be perceived as a task for society as a whole.

## 1. Introduction

### 1.1. Background

There is no longer any doubt that breastfeeding is the healthiest choice for newborn and infant nutrition [1]. Breastfeeding is associated with a reduction in mortality risk in both developing [2] and industrialized countries [3,4]. Breastfeeding reduces the long-term infant risk of type 2 diabetes mellitus and obesity [5] as well as asthma [6]. Breast milk provides several bioactive molecules that contribute to immune maturation, organ development and healthy gut microbial colonization, ensuring an appropriate immunological response that protects the newborn from infection and inflammation [7]. Breastfeeding is associated with a reduction in maternal breast [8,9] and ovarian cancer risk [10] and also reduces the risk of developing postpartum depression [11].

What is challenging are the targets set by the WHO to achieve exclusive breastfeeding in the first six months after birth. Many countries are struggling with these lofty expectations, which were previously “at least 50% by 2025” and have now been raised to “70% by 2030”. A deeper look at the global data reveals a surprising diversity. Zong et al. gave a detailed overview of the results of the Demographic and Health Surveys for 57 low- and middle-income countries (LMIC) from 2010 to 2018; the information on breastfeeding was collected in face-to-face interviews with a structured questionnaire interviewing the mother. The report shows that the highest prevalence of exclusive breastfeeding up to 6 months and continued breastfeeding up to 12 months is found in some African and Southeast Asian countries [12]. 

The UNICEF Global Database on Infant and Young Child Feeding, last updated in October 2023 [13], shows point estimates for the exclusive breastfeeding of infants aged 0–5 months, with the explanation of the point estimate being “the prevalence of the practice in the population”, so these figures should match those of Zong et al. [12]. This is only partially the case; for instance, the UNICEF database gives a prevalence for exclusive breastfeeding of 63.1% for Bangladesh [13], whereas in the study by Zong et al., it is 15% [12]. It remains unclear where these large differences result from. Lower breastfeeding rates are found in almost all European countries [14].

Breastfeeding rates in Germany are also low by European standards. There are two sources of data, both matching well to when the abovementioned numbers had been collected. The first is “Studie zur Gesundheit von Kindern und Jugendlichen in Deutschland”, abbreviated as KiGGS wave II with data from 2014 to 2017 [15], and the second is “Stillen und Säuglingsernährung in Deutschland”, abbreviated as SuSe II with data from 2017 to 2019 [16]. The data indicate that Germany performs well in terms of “any breastfeeding” at around 87% [15], but the rate of exclusive breastfeeding is very low. After 6 months, the rates end up between 8.3% [16] and 13% [15]. The changes in breastfeeding rates from birth to 6 months of age according to KiGGS and SuSe II are shown in Figure 1.

### 1.2. Aims

This leads to the question of why and what causes mothers, who have all possible legal support for maternity leave and time off work after the birth of their child [17,18,19], to stop exclusive breastfeeding so early. This is precisely the main concern of this study. It may be suggested that decisions about breastfeeding are made in the framework of complex social processes that need to be taken into account when identifying potential barriers [20]. There is an influence from the mother’s level of education but not from her health literacy [15,21]. The obvious first explanation regarding the data shown above is the question of how wealth influences the decision of whether to breastfeed or not. As soon as there is enough money available to buy formula and water quality allows its use, there seems to be a big temptation to do so. The same might be true for mothers who have to earn their keep and cannot afford the necessary time for breastfeeding, especially when they are left without a legal framework for maternity leave. 

We would therefore like to find out more about what it takes to convince mothers in favor of the healthier but more exhausting choice to breastfeed. 

## 2. Materials and Methods

### 2.1. Study Design

The purpose of this study was to first summarize the current state of research on breastfeeding barriers and enablers, including all relevant studies that have been published since the review carried out by Patil et al. (2020) [22] or that were not included in its scope. The research was organized as a systematic search and the following comments were performed as a narrative review. Narrative reviews usually provide a broad overview of a specific topic. They are therefore well suited for gaining an overview of the current state of research. Compared to systematic reviews, narrative reviews focus less on the methodological quality of the included studies to assess the risk of bias. The emphasis is on reflecting the breadth of the current state of research as comprehensively as possible. It is recommended that narrative reviews also use the PRISMA scheme in order to reduce subjectivity [23,24,25]. 

### 2.2. Literature Search: Inclusion and Exclusion Criteria

The literature search was conducted using PubMed on 19 March 2024. Studies published in the last 10 years were included if they were not considered by Patil et al. (2020) [22]. Studies published since January 2014, unless included in the previous review, were considered that dealt with barriers (social, cultural, environmental and workplace-related) and possible support options for breastfeeding in both healthy and sick newborns (preterm birth, NEC) in order to develop app-based support in the next step that precisely addresses these barriers (app development and evaluation are not included in this paper) and to derive concrete instructions for midwives. Only English-language studies from industrialized countries (Europe, USA, Australia, Korea, Canada, etc.) and some emerging countries (e.g., Thailand) were included; there was no restriction according to study design. Studies from developing countries were excluded. The Boolean operators AND and OR were used in the search in order to combine the terms in a meaningful way. The search was limited to a title/abstract search, which means that only studies in which all components of the search term appeared either in the title or in the abstract of the studies were considered in order to prevent studies with a different focus from being identified. The following search terms were combined: (breastfeeding OR breast milk OR mother’s/maternal milk) AND (barriers OR hurdles OR obstacles OR influencing factors OR support) AND (newborn OR preterm OR NEC). 

### 2.3. Literature Selection and Synthesis: PRISMA

After conducting the systematic search, the systematic literature analysis took place, which was based on the flowchart of the PRISMA protocol (PRISMA = Preferred Reporting Items for Systematic Reviews and Meta-Analyses), and was adapted according to the needs of a narrative review. 

The PRISMA statement is generally intended to help authors improve the reporting of systematic reviews and meta-analyses [26]. After the systematic literature search, duplicates were first removed, then the titles or abstracts of the remaining studies were analyzed (pre-selected) and unsuitable studies were excluded. The remaining titles were then assessed in full text and summarized qualitatively and quantitatively (Figure 2). 

Figure 2 shows that *n* = 139 studies were initially identified in PubMed, which were first screened with regard to the titles. Here, *n* = 85 studies were excluded since they did not fit the research question. Abstracts of *n* = 54 studies were then screened. Here again, *n* = 14 studies were excluded because they were duplicates, because the collectives were not representative of the conditions in Central Europe, or because of the language (study in Korean). Subsequently, *n* = 40 studies were checked for suitability in the full text, whereby *n* = 2 studies were again excluded because of methodological deficiencies. During the screening of the full texts, *n* = 13 studies were identified as potentially relevant from the bibliographies of the full-text screened studies and also reported in full text (Round 2), of which *n* = 3 were excluded again due to methodological deficiencies. Finally, *n* = 48 should have been taken into account as part of this review.

After re-reading the full texts, discrepancies were found in a review that summarized studies published since 2002, both in terms of methodology and results [27]. Subsequently, the studies included in this review were also organized and read, and it was found that the central statements of the studies in the review were in part incorrectly and arbitrarily reproduced [27]. For example, Carpay et al. (2021) stated that in an Icelandic study [28], women who were not in a relationship had a 6.63-fold higher odds ratio of not exclusively breastfeeding in the adjusted model but failed to mention that there was no significant effect after further adjustment [27,28]. Against this background, it was decided to include the *n* = 9 studies analyzed by Carpay et al. (2021), which were also suitable for the research question focused on here, even if they did not meet the initial inclusion and exclusion criteria (with regard to publication date), in order to reduce the risk of bias. Thus, *n* = 57 studies were considered and included in this review.

### 2.4. Structured Procedure

The research provided two publications introducing a model to group-identified arguments on barriers and enablers of breastfeeding: one from Dunn et al. (2015a) [29] and the other from Patil et al. (2020) [22]. Dunn referred to the Social Ecological Model and Patil used an adopted version of Hector’s framework. Both agree in distinguishing individual, environmental and society-related factors to influence breastfeeding behavior and regard aspects concerning support from the family and the community as well as hospital staff skills and workplace conditions as important within the group level. Interestingly, Patil does not mention legal regulation as a political element to organize society and rather refers to economic and educational issues within the society-related factors. Nevertheless, both frameworks have a lot in common, although the authors did not quote each other, which makes this model rather attractive.

In this paper, the authors also decided to base their findings on this classification but included social law, as this is of most importance for European healthcare performance. Concerning the identification of specific barriers or enablers, the two papers [22,29] choose very different approaches: Dunn et al. formed a coalition with 43 field-based professionals in New Hampshire, divided into six focus groups, and asked them five specified questions within semi-structured interviews [29]. Patil et al. performed a systematic review of barriers to exclusive breastfeeding based on literature research covering publications from January 1990 to October 2017 [22]. In the end, both approaches show remarkable similarities in their results, which will be addressed in the discussion. 

Nevertheless, here, the authors clearly voted to update Patil’s list of barriers but also consider the enablers of breastfeeding. As already explained, the literature research from 19 March 2024 included publications from the last ten years, so this paper had three years of overlap with Patil‘s research slot. The fact that some articles dating from these three years between 2014 and 2017 were included can be explained by the fact that the focus of this paper was not on exclusive breastfeeding, as performed by Patil et al. (2020) [22], but breastfeeding articles in general. In order to produce a result suitable to base probable support options in Germany on, the resulting list of barriers from Patil was extracted for the quoted literature from high-income countries alone. Within these, the publication by Wambach et al. from 2016 [30] was also removed, as the population studied comprised Mexican American women only, whose life conditions do not match European standards. The resulting body of evidence was created by enriching the list with all new items of proven significance from the 57 publications included in this review. 

### 2.5. Certainty of Statements

The main task of a systematic review is to reliably present the conclusions drawn from actual scientific research, i.e., to produce evidence. In order to base trustworthy decisions on these, the authors of this current paper followed the security advice of statements from the German Institut für Qualität und Wirtschaftlichkeit im Gesundheitswesen (IQWiG): Results have to be taken from publications with high methodological quality.If there is just one such result, it counts as a hint of evidence.For being trusted as proof of evidence, either a meta-analysis or two such results that do not contradict each other, thus pointing in the same direction of effect, are necessary [31].

The reader will identify these findings marked by bold characters in Tables 1 and 2 of the results. Aspects highlighted in blue can be understood as an extension of the state of research presented by Patil et al. (2020) [22].

## 3. Results

### 3.1. Characteristics of Included Studies

Appendix A shows the overall characteristics of the *n* = 57 included studies in relation to study design, aims, participants and interventions. A total of eight reviews, six observational studies, six cohort studies, seven studies with cross-sectional design, four RCTs, four quasi-experimental studies, four web-based surveys, two secondary data analyses (quantitative), eight studies with inductive and qualitative design, three studies with a mixed method approach and five other studies were included. 

### 3.2. Main Results of Included Studies 

Appendix A shows the main outcomes of the 57 included studies. Not all studies produced quantifiable results. In Section 4, the central results with reference to the research question were integrated into the model following Dunn et al. (2015a) [29] and Patil et al. (2020) [22].

#### 3.2.1. Healthy Mothers with Healthy Term Babies

Fortunately, there is a Cochrane review available to broadly cover this topic; McFadden et al. reported the resulting evidence for support of breastfeeding in an updated version from 2017 [32]. Other than this status quo report, which describes the situation before COVID-19, our research produced just two more articles within this category: one from China [33] and the other from Greece [34].

As breastfeeding rates are low, even in many high-income countries, McFadden et al. were interested in the effectiveness of different modes of offering similar supportive interventions and examined how and when these happened. The updated review includes 100 randomized controlled trials, of which 31 were new and 73 contributed to analyses. Evidence came from 29 countries and involved 74,656 women (63% from high-income countries) [32]. The outcome parameter for breastfeeding up to 6 months or up to 4–6 weeks differentiated between any and exclusive breastfeeding and measured whether the support given was able to reduce the risk of stopping breastfeeding. The summary of findings for all kinds of support versus usual care showed a significant effect in all four variants of measure. So, support does help, but it needs to be timely and skilled. The widespread lack of appropriate education for health professionals was a key factor [32]. Subgroup analyses by type of supporter, type of support and background initiation rate identified further conditions for successful interventions [32]. Although most studies involved professionals, support from non-professionals was associated with a broadly similar treatment effect to that from professionals, but we have to keep in mind that in 50 out of the 73 studies, people providing support had additional training to do so [32]. Concerning the type of support, face-to-face interventions were associated with greater effects. The results were similar for schedules containing an antenatal component compared to those offering postnatal support only [32]. The initiation rate turned out to be an important indicator for successful intervention: the higher the background rates, the greater the effect of preventing the stopping of breastfeeding early. In order to help mothers to keep up exclusive breastfeeding for six months, four to eight postnatal contacts performed best [32].

The publication from Zhang et al. refers to the theory of planned behavior and thus examines four factors that influence exclusive breastfeeding on an individual level [33]. These are knowledge, attitude, subjective norm (i.e., breastfeeding support from others in mother’s social network) and practice control (i.e., skill of breastfeeding). A total of 400 first-time mothers at 4 months postnatal who had given birth at Shanghai First Maternity and Infant Hospital (15,000 to 18,000 deliveries per year) were chosen to answer three questionnaires: one for demographic data collection, a modified breastfeeding attrition prediction tool and the breastfeeding knowledge scale. A total of 272 completed questionnaires were returned. The rate of exclusive breastfeeding at 4 months was 34.4%. Mothers with higher incomes were less likely to practice exclusive breastfeeding. All the four relevant parameters of the theory of planned behavior turned out to be significantly involved in successful exclusive breastfeeding but the most influential variable was knowledge [33].

The third and last publication to be mentioned in this section of the review exactly addresses this option: Iliadou et al. describe the effect of a midwife-led education program at the tertiary hospital in Athens, Greece [34]. A total of 203 nulliparous pregnant women > 32 weeks of gestation participated in a quasi-experimental study. The intervention group followed a four-hour midwife-led antenatal tuition program, covering several topics around breastfeeding such as anatomy and physiology of milk production, skills for latching and positioning and also addressing attitudes and self-efficacy as well as offering to build a support network for the members of the intervention group [34]. Four questionnaires were used to measure the intervention effect: the Breastfeeding Self-Efficacy Scale, the Iowa Infant Feeding Attitude Scale, the Breastfeeding Knowledge Questionnaire and the Perceived Breast Feeding Barriers Questionnaire, all translated into Greek [34]. Attitudes towards breastfeeding, knowledge about breastfeeding and breastfeeding self-efficacy were all significantly improved among women in the intervention group but were unchanged in the control group. The perceived barrier score dropped markedly in the intervention group and raised hope that the intervention might have altered thoughts. Unfortunately, no postnatal breastfeeding rates are reported. It would have been interesting to find out whether or not the intervention could raise the amount of 0.7% only given as national data for Greece in the introduction of the paper [34].

#### 3.2.2. Impaired Maternal Health

German textbooks of midwifery list absolute as well as relative maternal contraindications for breastfeeding, such as certain infectious diseases and severe complications postpartum that afford intensive therapy, oncologic treatment or hard drug dependency; in these cases, breastfeeding is impossible. Mothers with deformities of their breasts or those who have undergone breast surgery may have difficulties, as may neurological (e.g., epileptic or multiple sclerosis) or psychiatric patients. In these circumstances, medical treatment is needed and a decision has to be taken if and when breastmilk feeding can be achieved or not [35,36]. Though all these medical reasons seem much more relevant, surprisingly, our research detected four publications that mention maternal obesity as a breastfeeding barrier [37,38,39,40]. The more one considers this point, the clearer it proves not to be a coincidence—obesity is indeed a large and widespread problem, with 121,239 out of 707,621 mothers showing a BMI > 30 in 2022 in Germany (17.1%) [41]. According to the results of the four studies [37,38,39,40], breastfeeding can be described as a way for mothers who are overweight or obese to lose weight. The prerequisite is that breastfeeding is not given up, and that is exactly what these four articles examined.

Kozhimannil et al. investigated early breastfeeding in the context of complex pregnancies, i.e., clinical conditions either before entering pregnancy or induced by pregnancy in the USA [37]. Keely et al. set about to explore factors that influence breastfeeding practices in obese women who either stopped breastfeeding or were no longer breastfeeding 6–10 weeks after birth in the UK [38]. L. Kair and T. Colaizy examined data from the Centers for Disease Control and Prevention Pregnancy Risk Assessment Monitoring System (PRAMS) from Illinois, Maine and Vermont in the USA [39], and Mallan et al. compared self-reported breastfeeding problems in non-overweight and overweight women in Australia [40].

Kozhimannil et al. used data from 2400 women who gave birth in a US hospital to examine the relationship between pregnancy complexity and early infant feeding 1 week postpartum. More than 33% of women had a complex pregnancy (taking hypertension, diabetes and obesity as index diagnoses). These women had 30% lower odds of intending to breastfeed. Those who intended to breastfeed had similar rates of any breastfeeding 1 week postpartum, but complexity was associated with >30% lower odds of exclusive breastfeeding. High levels of hospital support for breastfeeding were associated with nearly three times the odds of exclusive breastfeeding 1 week postpartum, which shows the importance of early encouragement, i.e., when the mother and child are still in hospital [37].

Keely et al. came closest to the underlying problems by performing face-to-face interviews with 28 obese women at 6–10 weeks after birth [38]. The authors identified three major themes: -The impact of birth complications: only seven had a spontaneous vaginal delivery, two had a forceps delivery and 19 of 28 women gave birth by emergency cesarean section.-A lack of privacy: busy wards as well as too many visitors at home with mothers feeling reluctant to breastfeed in front of them.-A low uptake of specialist breastfeeding support (which in the UK means attending a clinic, since midwife home visits do not exist) [38]. 

PRAMS data of 6467 women showed significant differences in demographic characteristics in four defined BMI categories; out of the 312 cases with maternal diabetes, 226 (72.4%) were either overweight or obese. Adjusted Odds Ratios for barriers to breastfeeding continuation were significant for perceived insufficient milk supply not satisfying the infant, breastfeeding difficulties or infant jaundice. Since answers had been set in this survey, the instrument could neither provide an explanation of what mothers fed their babies who seemed dissatisfied with breast milk alone nor explain specific difficulties [39].

Mallan et al. could not find significant differences in breastfeeding problems in the first month post-partum between 303 non-overweight and 159 overweight women, especially not for the feature “not enough milk“ [40]. Previous work is cited that has indicated that perceptions of insufficient milk are consistently associated with low maternal self-efficacy, which matches well with the finding of this present study with almost double the proportion of overweight mothers who agreed that an important reason for using formula was that it was as good as breastmilk. Nevertheless, the true reason for giving formula seems unclear and again shows the shortcomings of questionnaires together with high attrition between baseline and follow-up participants in the final sample of this study [40].

*Comment:* Giving birth warrants not only a lot of patience from both the mother and obstetrician but is much easier for women who are slim and fit. Obesity often causes birth problems, long and exhausting hours in labor as well as forceps or cesarean section for delivery. These surgical interventions can decrease early skin-to-skin contact or make breastfeeding otherwise difficult (by not allowing for a comfortable feeding position for instance). On top of these obvious problems, there are other issues to be solved: the striking difference between the objective finding of as much milk in overweight women compared to non-overweight women and the maternal perceptions of unsatisfied breastfed infants might find an explication in the higher prevalence of diabetes in the overweight mothers—these babies are used to high intrauterine glucose levels and could indeed seem hungry on breastmilk alone. German legislation does allow for coverage for up to 12 weeks after birth if needed, and even longer if prescribed by a doctor. Midwives not only give valuable information and practical help to the woman but also monitor infant well-being, detecting growth problems or jaundice early and constantly reassuring young mothers to cope with any kind of breastfeeding issue [32,33,34].

#### 3.2.3. Sick Children, Preterms and Twins

Acute or chronic illness, disability or congenital anomaly—all these conditions can make breastfeeding even more difficult. They generally cause hypotonia, somnolence and poor weight gain in a cluster. There is a tendency for poor and weak suckling and uncoordinated suck–swallow–breathe reflex sequences, not only in children with obvious obstacles to produce this sophisticated coordination necessary for breastfeeding (such as macroglossia, small intra-oral space or cleft lips) but also in those with a congenital heart problem or Down syndrome. Mothers not only need even more patience with frequent short feeds but also have to adopt an attitude of constant flexibility [3,7].

There are indications that breastfeeding rates are generally lower in preterms compared to babies born at term, which makes prematurity a risk factor for inadequate breastfeeding, although premature babies benefit greatly (e.g., in terms of the risk of necrotizing enterocolitis (NEC)), and therefore, an increase in the breastfeeding rate should be aimed for here in particular [3,7]. Studies by Carpay et al. (2021) [27], Jónsdóttir et al. (2020) [28] and others point to lower breastfeeding rates in preterm infants. Jónsdóttir et al. (2020) showed that the breastfeeding rate in late preterm infants was lower than in full-term infants, regardless of whether they were cared for in the NICU or in a normal ward [28]. Rayfield et al. (2015) were also able to demonstrate a significantly lower breastfeeding rate for preterm infants in a longitudinal study (10 days and 6 weeks postpartum) [42]. Goyal et al. (2014) compared late preterm infants (34 to 36 weeks) with early term infants (37 to 38 weeks) and term infants (39 to 41 weeks) and were able to show that both the rate of any style of and exclusive breastfeeding after 1 week postpartum increased proportionally with the increase in gestational age, i.e., were the lowest in the group of late preterms [43]. Demirci et al. (2013) also demonstrated in a cohort study of more than 1 million mothers that mothers of 35- and 36-week infants were slightly but significantly (*p* < 0.01) more likely to initiate breastfeeding than mothers of 34-week infants [44]. Twin premature babies are particularly rarely breastfed [45], so they must be regarded as a special risk group that requires special support. 

*Barriers*: A total of seven studies looked at the obstacles to breastfeeding in preterm infants. The regression model of Demirci et al. (2013) indicated that interactions involving sociodemographic variables, including marital status, age, race/ethnicity, education, parity and smoking status were among the most significant factors associated with breastfeeding non-initiation (*p* = <0.05) [44]. Crippa et al. (2019) were able to demonstrate in an Italian study that maternal age over 35 years, Italian ethnicity, the feeling of reduced milk supply and twin pregnancies were risk factors for the early cessation of breastfeeding [45]. Early feeding with artificial supplementary milk must also be described as a central hurdle. Late preterm infants who were regularly fed with artificial supplementary milk had to wait longer for breastfeeding to begin and were breastfed less frequently, according to the results of a study from Sweden [46]. The study by Kair and Colaizy (2016b), which also looked at barriers to breastfeeding in preterm infants included a total of 2530 mothers of late preterms who were able to breastfeed either in the NICU or in a nursery room [47]. 

Regardless of NICU admission, the top reasons cited by mothers for early breastfeeding discontinuation were perceived inadequate milk supply and nursing difficulties [47]. Yang et al. (2019) also conducted a qualitative survey with Chinese women whose preterm babies were cared for separately from them in the NICU. They reported physically and psychologically difficult breastfeeding experiences during the time they were separated from their babies. They considered expressing breast milk to be an integral part of their maternal role, even though some found expressing breast milk exhausting. They also cited a lack of confidence in the quality of breast milk as a further obstacle [48]. Study results from Italy also suggest that the need to pump and inadequate support are among the key factors why premature babies do not receive breast milk [48,49]. In a study of 92 mothers of 121 preterm infants, Gianni et al. (2016) found that one day before discharge from the hospital, some women still had problems with latching and milk supply [49]. Gianni et al. (2018) found in a cohort of 64 mothers of 81 preterm infants that problems with pumping increased the OR for exclusive formula feeding by 4.6 times and problems with milk volume were associated with an OR of 3.57 for weaning—both also an indicator of inadequate training or support in hospital [50]. Gianni et al. (2016) enrolled mothers of late preterms (gestational age of 34 0/7 to 36 6/7 weeks) [49] and Gianni et al. (2018) included mothers of preterms with a gestational age ≤ 33 weeks [50].

The hospital environment must therefore also be considered a key risk factor. If preterm infants receive formula at an early stage in the maternity hospital, it is highly likely that breastfeeding success or breastfeeding practices are inadequate, which represents a significant health risk, especially for preterm infants [46]. It has also been shown that the common practice (e.g., in Chinese hospitals) of separating premature babies and mothers in the NICU has a negative impact on breastfeeding success, according to the results of Yang et al. (2019) [48]. Consequently, subsequent breastfeeding success is also negatively influenced if there is a temporary separation of mother and child due to transfer, as indicated in an Italian study (Gianni et al., 2016) [49], or if the hospital apparently offers insufficient breastfeeding support [48,49]. The systematic review by Hookway et al. (2021) also looked at the hospital setting as a barrier to breastfeeding, albeit with a focus on sick children and not explicitly premature infants [51]. Their findings point to practical and psychological challenges in continuing breastfeeding in hospital, complications of the illness that make breastfeeding difficult, lack of specialist breastfeeding support from hospital staff and the lack of availability of specialist equipment to support complex breastfeeding. The results confirm the lack of consistent, high-quality care for breastfeeding support in pediatrics [51]. 

*Enablers*: Rayfield et al. (2015) showed that mothers who reported receiving contact details for breastfeeding support groups had a higher likelihood of breastfeeding late preterm infants after discharge [42]. Goyal et al. (2014) were able to show that high hospital support significantly increases the breastfeeding rate, although this is offered significantly less frequently to mothers with preterms in the USA. Just 16.4 percent of late preterm infants experienced such support, compared with early-term (37.9%) and term (30.7%) infants (*p* = 0.004) [43]. 

Crippa et al. (2019) point out the central importance of maternal education as a breastfeeding-promoting factor [45], also confirmed by Kair and Colaizy (2016a) [39] and Kair and Colaizy (2016b) [47]. According to the results of a Swedish study (Gerhardsson et al., 2018), maternal self-efficacy is an important predictor of breastfeeding duration in late preterm infants [52], which points to the central function of midwives to promote the self-efficacy of pregnant women and women who have recently given birth in the context of the salutogenesis principle. Several studies have looked at the effect of Kangaroo Care on the breastfeeding rate of premature babies: Hake-Brooks and Anderson (2008) were able to show in an RCT with 66 mother–child pairs that there was a significantly longer breastfeeding period with Kangaroo Care compared to the control group (5.08 months vs. 2.05 months with *p* = 0.003), and the frequency analysis (although not analyzed for significance) showed higher breastfeeding rates in the Kangaroo Care group at all time points (at discharge and after 1.5 months, 3 months, 6 months, 12 months and 18 months) [53]. Mörelius et al. (2015) also investigated the effect of skin-to-skin care in preterm infants compared to standard care with regard to breastfeeding rates as part of an RCT. The frequency analysis also showed higher breastfeeding rates with skin-to-skin care at all three observation points (on discharge from hospital, after 1 month and after 4 months), but the differences were not significant due to the small number of cases [54]. Gianni et al. (2016) were also able to demonstrate a higher breastfeeding rate with Kangaroo Care after premature birth [49]. The positive effect of Kangaroo Care on premature babies was also shown in a French study with 883 premature babies born in the 32nd to 34th weeks of gestation. Kangaroo Care increased the OR for breast milk feeding by 2.03 times [55]. Mitha et al. (2019) further demonstrated that the OR for breastmilk feeding is also increased with early parental involvement in feeding support (OR = 1.94, 95% CI: 1.23–3.04) and when the ward is trained in a neurodevelopmental care program (OR = 2.57, 95% CI: 1.18–5.60), as well as in parents from regions where a high level of knowledge about the benefits of breastfeeding is prevalent in the general population [55].

Finally, some studies also looked at the effect of maternal education programs or support that extends beyond the hospital stay. Niela-Vilén et al. (2016) investigated whether an Internet-based peer-support intervention via social media has an effect on the duration of breastfeeding or breast milk expression or maternal breastfeeding attitude [56]. Mothers with relevant previous experience and a midwife were asked to support the study participants in a Facebook group for breastfeeding difficulties 1 year after birth, but the intervention had no significant effect on the duration of breastfeeding [56]. Ericson et al. (2018) were also unable to show a significant influence on the likelihood of exclusive breastfeeding in their intervention. In this study, *n* = 493 mothers of preterm infants were included. Participants in the intervention group received a call every day to provide support in the event of breastfeeding difficulties, while participants in the control group had to report any problems themselves [57]. The quality of telephone counseling in the study [58] was examined by Ericson and Palmér (2018) in another study. Here, the mothers reported how they were empowered by the telephone support, and that they were listened to and treated with respect, understanding and knowledge. The support was individually tailored and included both practical and emotional support. At the same time, the mothers indicated that the quality of the advice was very dependent on the individual professional [58]. 

Other maternal support programs were more successful: Jang and Hong (2020) found a 5.18-fold higher OR for exclusive breastfeeding for mothers in the intervention group than in the control group in their quasi-experimental study with 40 included mothers of late preterm infants. The intervention group received a web-based breastfeeding education program as well as practical support through home visits for mothers over 4 weeks [59]. In the study by Estalella et al. (2020), there were significantly more women in the intervention group who exclusively breastfed their preterm infants than women in the control group (68.4% vs. 50.7%, *p* = 0.002). Women in the intervention group were 2.66 times more likely to use a breast pump. However, due to methodological weaknesses (including no intention-to-treat evaluation), the generalizability of the results remains questionable [60]. In a study from Wuhan in China, a WeChat program for online instruction and support over 3 months significantly increased the study participants’ knowledge of breastfeeding, but there was no effect on the breastfeeding rate [61].

Kuhnly (2015) [62] described, in her case study, a whole range of interventions to support breastfeeding in a mother with premature twins. Strategies to support sustained breastfeeding in late preterm multiple-birth infants include developing a family-centered feeding plan in collaboration with the medical team, assessing and supporting breastfeeding sessions, promoting lactogenesis with pumping or manual expression and activating a support system for families [62].

*Professional education*: If it comes to hospitalization, things often grow worse due to the separation of mother and child as well as due to ignorance about breastfeeding in general by the staff. Our research produced eight publications addressing this very problem. Hookway and Brown (2023) investigated resulting barriers to optimal breastfeeding in the UK pediatric setting by a large self-reported survey, i.e., they raised 409 professionals, mainly nurses and doctors to answer a questionnaire [63]. The study explored the barriers experienced by staff as well as professionals‘ perceptions of parent barriers. The more experienced a professional was, the fewer barriers were identified. There was a general lack of knowledge, including about the breastfeeding policy of the hospital, with nurses having a greater awareness than pediatricians. In the UK, the Baby-Friendly Hospital Initiative does not cover pediatrics. Staff priority for accurate fluid balance management and conflicting information together with practical problems (inability to find a breast pump on the ward) are mentioned, and it seems to be coincidental if parents find adequate breastfeeding support. Often, bottle feeding with formula is regarded to be easier, as it increases weight gain more rapidly and brings families home quicker. The authors conclude that pediatric healthcare professionals urgently need sufficient skills and knowledge about breastfeeding [61]. The same underlying shortcomings have been identified in two other papers, with Michaud-Létourneau et al. (2022) starting to develop a comprehensive undergraduate training program for various healthcare professionals in Quebec/Canada [64] and del Valle Ramirez-Duran et al. (2024) already informing about a very successful intervention on the same purpose, in this case with a quasi-experimentally designed teaching study on an established educational program for nurses and midwives in Avila, Spain [65]. While the Canadian study was already aware of the problem and initiated the Quebec Breastfeeding Movement, they took a broader scope for their enterprise and aimed to reach the workforce in healthcare as well as in the educational system in the hope of establishing a professional network with the same level of knowledge, giving helpful and uniform information about breastfeeding to parents [64]. The Spanish initiative is more pragmatic and trains potential supporters by testing a new course format for student’s education. A total of 40 volunteers were separated into two groups, with one experiencing an enriched setting with focus groups, clinical simulation and a visit to the local breastfeeding association and the other being the standard teaching. In the end, a validated questionnaire showed a significant difference in knowledge between the two groups. Students were especially pleased to find themselves in role-play, simulating communication challenges they felt unprepared for [65]. The results of Ericson and Palmér (2018) also point to the need for training for healthcare staff. Some of the mothers surveyed felt negatively affected by controlling and intrusive healthcare staff and stated that the provision of support was dependent on the respective healthcare professional and that they therefore felt “like they were in a lottery” [58]. The need for training of healthcare staff is not least a result of the studies presented above, which dealt with the barriers of the hospital environment [48,49,50,51]. 

*Comment*: Separation between mother and child has been proven to be very harmful for breastfeeding; nevertheless, it happens regularly when sick children are in pediatric hospital care [47,48]. Breastfeeding is then often perceived as a hindrance to professional procedures, as it is time-consuming and brings parents’ wishes into direct confrontation with ward culture. Many pediatric units as well as nurseries have an uncritical attitude towards formula use, which makes things even worse. It is obviously not a coincidence that the most comprehensive list of obstetric units in German hospitals is held by a company producing formula, not by a state-run institution [33]. The general improvement of knowledge and skills around breastfeeding is badly needed; breastfeeding should be given more attention in training programs for healthcare students of any profession involved. To address this, WHO and UNICEF (the United Nations Children’s Fund) have not only developed breastfeeding training programs but also published a Competency Verification Toolkit to help ensure that healthcare providers possess breastfeeding competencies [66]. A wide range of research findings point to the need for appropriate measures to enable premature babies to be breastfed [3,7]. Already in 1999, Schanler et al. showed that preterm infants fed with breast milk could be discharged earlier (73 ± 19 vs. 88 ± 47 days) compared to preterm infants fed with formula, although they gained weight significantly more slowly (22 ± 7 vs. 26 ± 6 g kg^−1^·day^−1^). The incidence of NEC and late-onset sepsis was also significantly lower in the breast milk group [67]. 

#### 3.2.4. Workplace Conditions

Whether or not to return to work after birth confronts mothers with a profound decision concerning their future lifestyle: it will always be a challenge to balance their family attitude with the demands of paid work, and any decision has to be respected. No doubt many women have to earn their living and are therefore certain from the very beginning, but it is often difficult to combine breastfeeding with the demands of paid work. Dunn et al. (2015b) found that breastfeeding duration was significantly related to employment status; among women who breastfed for 6 months or longer, 15% were employed full-time, 30% worked part-time, and 55% indicated “other”, such as unemployed or stay-at-home mother (*p* = 0.01) [68]. But for all the in-between situations where mothers manage to combine breastfeeding and paid work to have a healthier child in the future, it is worth exploring their experiences and finding out what conditions can enable them to succeed. Two of the publications that were identified to cover this aspect treat workplace barriers in a more general way and describe the results of qualitative interviews—one from the USA [69], and the other from Ireland [70]. 

Fourteen mothers participated either in a focus group, interviews or both, as well as 17 senior business managers in rural Missouri in the first paper [69], and 16 women were interviewed in the second [70]. Both authors performed a content analysis of the transcripts in order to select prevalent themes. The Irish distinguished “Culture”, “Support”, “Information provision”, “Returning to work” and “Feeding in the Workplace” whereas the Americans found “Tolerance”, “Flexibility” and “Pro-activeness” to frame their findings. Though the Irish paper starts from a broader context to describe barriers (such as breastfeeding in public for instance) [70] and the American paper pragmatically concentrates on the workplace situation [69], both identify the same problems in the end. In Ireland, paid maternity leave is 26 weeks and mothers who wish to exclusively breastfeed for 6 months or longer find themselves having to take holiday leave or unpaid leave from work in order to meet the WHO guidelines [70]. Hospital stay turned out to be the most difficult time in the breastfeeding experience for most of the participants due to lack of support and lack of knowledge from healthcare professionals. Women describing themselves as “stubborn” or “determined” discovered that this feature helped to overcome any obstacles they faced. Return to work warranted an established feeding routine, i.e., the newborn had to accept being bottle-fed; this caused emotional stress and anxiety in many women because it had to be accomplished on time. All women described difficulties with lack of facilities to express and store breast milk at work; many did not disclose to their employers that they were breastfeeding [69,70].

The American authors were quite upset about the fact that four years after the passage of the Affordable Care Act, there were considerable shortcomings in compliance with the new law: inadequate breastfeeding information for mothers, lack of support from co-workers and supervisors, issues concerning the amount and duration of breaks allowed to express milk as well as inadequate locations provided. In the conclusion, the absence of established infrastructure and policy in the workplace regarding milk expression turned out to be an essential failure to meet the legal standard [69].

Do these findings differ in another line of business? What does the situation look like in the US healthcare sector, where information on the positive health outcomes of breastfeeding should be widespread and influence flexibility?

Nine papers answer this question, the details of eight of them are given in Appendix A, whereby the three publications [71,72,73] were identified from the mini-review by Whiteside et al. (2020) [74]. The results from 3357 participants contacted in surveys and websites from 178 accredited orthopedic surgery residency programs screened showed a remarkable lack of progress towards helpful conditions for breastfeeding. Many mothers failed their personal goals, and almost all complained of inadequate time to pump, rigid schedules and insufficient space. Nevertheless, there is a marked difference by specialty. In emergency pediatrics, support is much better than in orthopedic surgery, as pediatric colleagues know better about breastfeeding and value the needs of the mother and child by professional orientation [71,72,73,74,75,76,77,78,79].

The last of these nine papers by Haas et al. (2020) is not a study, but a general overview about lactation in US hospitals and confronts the reader with a concept of “best practice”: a technical solution that allows the mother, working as an emergency physician, to continue to work clinically while pumping [80]. The wearable pump is battery-operated and contains the pump motor as well as milk-collection bottles in a breast-shaped device. “Although relatively discreet and quiet, wearable breast pumps may be visible or audible to others when in use”. This quotation from the original text shows how very different the topic of breastfeeding at work is regarded in the US: as if it was something to feel ashamed of, that needs to be hidden, that induces “guilt and sense of duty, that makes it challenging to leave for pumping”. This is not at all compatible with the idea of respect for the lactating mother, who needs privacy and peace. 

A surprisingly positive experience was published from a university hospital in Thailand in 2023. This cross-sectional study enrolled 110 mothers from 153 hospital employees (mostly nurses) who had taken maternity leave within the last 2 years [81]. The rate of exclusive breastfeeding for 6 months was 63.6% in this population. Multiple logistic regression showed “perception of breastfeeding obstacles“, “breastfeeding behavior“ and “support from health care system“ to be the significant factors with successful exclusive breastfeeding [81]. In Thailand, paid maternity leave is 45 days and can be extended to 90 days without pay for the latter half. Most of the study participants (i.e., more than 90% in each group) managed to take maternity leave for 90 days or more, irrespective of whether they exclusively breastfed or not. The same is found for other baseline characteristics (e.g., percentage of separation from infants), but the authors do not reveal exactly which measure made the difference to achieve these results. Maybe it is the breastfeeding experience and the prenatal breastfeeding education that show an (insignificant) difference between the two groups. Nevertheless, the nurses had three wishes for further support: clean refrigerators and designated space to express milk, official breastfeeding break time and postponing the night shift [81]. 

A promising approach can be derived from a systematic review that studied enablers and barriers to workplace breastfeeding in the Armed Forces [82]. The author not only found 16 military breastfeeding studies from high-income countries in English, but also listed 16 breastfeeding policy and guidance documents. The main message is that “promoting extended maternity leave for breastfeeding fuels the misconception that is it unacceptable in the workplace. To be effective, policies must empower the servicewomen and must be widely available”. As the Army is a well-organized enterprise with hierarchical responsibilities, occupational medicine gets a chance to be installed and thus permits these guidelines to be followed. In order to no longer depend on individual commander’s approaches concerning breastfeeding expectations and provisions, five key recommendations are given [82]:An easily accessible breastfeeding-specific policy is required.Education on breastfeeding by healthcare professionals has to be installed; awareness is a responsibility, as it is a maternal duty to inform her employer and medical officer of breastfeeding status.Individual risk assessment and breastfeeding plans are necessary, i.e., protection of breastfeeding servicewomen from exposure to any harmful occupational hazard.Policy must clarify minimum breastfeeding facility standards in all settings, i.e., in the home base as well as in exercise or on deployment.Define exemptions from deployment, release recommendations on physical activity and fitness testing [82].

The last article studied shows the consequences if such regulation is not installed [83]: it tells the story of women from the Chinese community in Madrid, who regularly refused breastfeeding in hospital. What first appeared to be triggered by cultural reasons turned out to be a severe cluster of adverse conditions: these young Chinese women mostly worked in a family enterprise, a restaurant, a shop or a clothing manufacturer. Not all family members usually lived in Spain, but those who did were expected to work in businesses where every hand is needed. These young mothers often sent their newborns home to be raised by their grandparents and they were consequently bottle-fed. Neither European law is applicable, nor does their employer inform them about legal regulations, as they do not want to lose the workforce in their business. The mothers therefore immediately returned to work and called their children “satellite babies“ [83].

*Comment*: Even industrialized countries are lightyears away from regarding breastfeeding women in the workplace as a normality. In order to ameliorate the situation, legal regulation and guidance for employers have to be installed; this seems to be easier in large enterprises that can afford a healthcare professional, i.e., someone responsible in making occupational medicine happen, who not only creates facilities but also assures access to these. This would be someone who would never put a breastfeeding nurse on a night shift or let these mothers work in otherwise dangerous environments.

#### 3.2.5. Community Initiatives

Since both publications identified under this headline refer to interventions within the US healthcare setting [84,85], the authors decided to include four further articles from US authors here and thus make this section a reflection on US breastfeeding support conditions, i.e., a situation that has to cope without a general social security standard [68,86,87,88].

First, some facts about breastfeeding in the US were taken from Sriraman and Kellams (2016): CDC data from 2014 specify the rate of ever breastfeeding to be 79.2%, any breastfeeding up to 6 months to be 49.4% and exclusive breastfeeding to be 18.8% [86]. Most women make up their minds about breastfeeding before conception or during pregnancy and those with lower confidence are more likely to stop within 1 week postpartum. Gynecologists as well as pediatricians feel that they have little or no breastfeeding education or training. Grandmothers often belong to a generation who did not breastfeed and are therefore unable to help. Many young mothers receive formula in maternity discharge packs or at home and the Women, Infants and Children Program (WIC) that serves the poor purchases over 50% of the formula in the US [86].

The above-quoted figures are confirmed by the cross-sectionally designed study reported by Dunn et al. (2015b) for WIC mothers in New Hampshire [68]: 220 out of 283 women (78%) initiated breastfeeding and 13% did so exclusively for at least 6 months. The study added some further information from the answers to a questionnaire with 65 questions. All of the following variables proved to be significant: educational level, non-white race and having planned to breastfeed prior to birth as predictors of breastfeeding initiation. There were significant differences in beliefs about breastfeeding between women who had ever breastfed and those who had never breastfed concerning losing baby weight, the health of breastfed babies and bonding. Maternal age turned out to be the most important for the duration of breastfeeding [68]. As long as the mother and child are in the hospital, the Baby-Friendly Hospital Initiative produces a helpful effect and does increase breastfeeding initiation, exclusivity and duration—even with an average stay of 1.6 days after delivery [68]. But what happens after discharge?

Sayres and Visentin (2018) see pediatricians to be the first providers for postpartum care and recommend that they network with nurses and other home healthcare providers in order to support women who struggle with the many barriers of breastfeeding [87]. The article also mentions the option of using mobile applications for support by quoting some studies. All these choices have to be paid out-of-pocket in the US system [87]. When it comes to maternity leave regulation, the ACOG’s (American College of Obstetricians and Gynecologists) contribution describes the situation after the Patient Protection and Affordable Care Act (ACA): despite the legal requirements, many workers are left without adequate protection, as it is up to the employers, with fewer than 50 employees, for example, to apply for an exception. In addition, many workers are exempt from this law and have to directly negotiate with their employers if they wish to pump milk at the workplace [88]. Consequently, young mothers are left alone with their breastfeeding enterprise. The US system cannot rely on social security regulations in charge of healthcare for this population because freelance midwives who work out-of-hospital and do home visits are not covered.

Help has to be organized within the community, i.e., relying on the neighborhood or other institutions to take over cost. Our research discovered just two initiatives of this kind, both regional and in a state of preliminary exploration or pilot projects [84,85].

The first describes the organizational preparation to support breastfeeding mothers of late preterm infants in the Mountain West region of the United States. In this location, the main source of breastfeeding support after hospital discharge is offered by peer counselors (i.e., lay persons) from the WIC program. The concept is based on the idea of establishing contact with these counselors prior to discharge and outlines the co-operational process of how to bring people together. Unfortunately, there are no breastfeeding outcome data [84]. The other initiative takes place in the historically African American community of Newtown, close to Sarasota, Florida [85]. A student’s project examined region-specific barriers to prolonged breastfeeding. A community breastfeeding conference was organized and attendees’ knowledge and confidence were assessed by a conference pre- and post-test. A total of 28 completed surveys were returned from this sample, and 22 of the respondents were of Caucasian or Hispanic ethnicity. Half of the mothers said that the lack of access to paid maternity was a barrier and the students had to acknowledge that this was a much more important fact than they had supposed. There was no deficit in maternal knowledge or motivation for breastfeeding [85]. Other than the fact that no obstetric outcome data had been asked for, it turned out to be of crucial relevance that only 50% of those who registered online for the conference actually attended; the authors suspected that holding the meeting during a weekday morning during spring holidays might have likely proved to be an obstacle to attendance [85].

### 3.3. Study Outcome: Barriers and Enablers to Breastfeeding—Synopsis from the Literature 

The resulting lists of barriers and enablers to breastfeeding as a synopsis of the literature in extension of the model by Patil et al. (2020) [22] are shown in Table 1 and Table 2. There is a considerably longer list for the hurdles (Table 1) compared to the enablers (Table 2), showing the enormous amount of difficulties mentioned in the body of evidence. But there is not only challenge in this finding, a lot of initiatives have been identified to bring support for stressed young mothers too. Compared to the initial list of barriers and enablers compiled by Patil and colleagues, there are some new themes, such as complex pregnancies or maternal obesity and preterms as well as issues that had been substantially extended, such as workplace-related problems.

For the sake of a clear conclusion, the authors decided to concentrate their comments regarding the results in this section on the conditions that offer access to intervention on an individual level, which means that maternal age, for example, will not be discussed here, as it cannot be changed. Sociodemographic items will also not be treated in detail, as it turned out that these are still very difficult to interpret. Demirci et al. (2013) made an interesting observation when calculating combinations of features within her data set from Pennsylvania [44]: she discovered that while single characteristics such as being married or having achieved higher grades of education do have a positive impact on breastfeeding initiation, their interaction when combined act negatively. Demirci concludes that these variables “may be proxy factors for employment, income, and additional home/childcare responsibilities”, which we know nothing of [44]. 

Workplace-related issues will be left for the discussion section since these do need to be reflected on the background of related social security systems, i.e., policy. The main resulting option for support is the transfer of all kinds of knowledge, be it to the mother or to the staff in hospital. Several publications have proven the fact that increased knowledge does help, either in theory or in practical support (Mitha et al. (2019) [55] or Estalella et al. (2020) [60], for example). More and more digital education appears promising in this field. Better knowledge of breastfeeding stimulation for hospital staff is also a very good investment in order to achieve better breastfeeding rates through cooperation between professionals and family members on the ward. Mitha et al. mention implementing a sensory–motor program for NICUs in France, as it integrates knowledge about neurological development into the necessary skills of how to handle a preterm newborn.

## 4. Discussion

### 4.1. Content-Related Discussion

This chapter invites a comparison of our results with those from the three other enterprises the authors found during their research; these are the two systematic reviews by Patil et al. (2020) [22] and Carpay et al. (2021) [27], as well as the earlier qualitative assessment conducted on barriers and positive contributors to breastfeeding by Dunn et al. (2015a) [29]. Even though all four enterprises differed in some aspects from each other, all active authors concluded that providing parents with more information and healthcare support is effective in improving the rates of breastfeeding exclusivity and duration. It is worth mentioning that Dunn’s results from semi-structured interviews with field-based professionals fell short of the issues “staff training” and “workplace problems” in their list of proposals for improvement [29]. Therefore, systematic literature research does identify more aspects to consider. Nevertheless, hits in research did not produce more than three double-matches between our approach and the one Carpay et al. undertook, although both had been performed in PubMed. Patil et al. used MEDLINE for research and found different articles. Even though the focus was slightly different, this finding does raise the question of which literature is seen by which research machine and which is neglected. Two really important contributions from the Cochrane Library have not been identified at all [89,90]: one from Quigley, Embleton and McGuire (2018), which shows that giving formula to preterms is associated with a higher risk of developing NEC, a severe gut disorder [89], and one from Brown, Walsh and McGuire (2019), who could not identify any RCT to compare formula versus maternal breast milk for feeding preterms or low-birth-weight infants [90]. Both concern the use of formula and shed either a rather negative light on its effect or voluntarily avoid such a result [89,90].

The second issue for discussion is the problem of realization, i.e., making things happen according to knowledge, which is not at all a natural consequence. As Rykiel et al. (2023) [85] pointed out in their survey response considerations, *“maternal knowledge and motivation were not a deficit in promoting exclusive breastfeeding”*, this was the conditions of the US healthcare system concerning maternal leave regulation even after the Affordable Care Act [85]. There are huge differences between the US situation and the European understanding of social security systems. All the barriers identified by Chiang et al. (2019) describe social disparity issues, which are not overcome by social law or social insurance in the US. Thus, without such a network to support the poor, help has to be organized by the community [91]. Strang and Broeks (2017) gave an overview of European maternity leave policies under the EU Maternity Leave Directive (92/85/EEC) and described their impact on breastfeeding practices as well as on labor market demands [92]. Of course, there are different regulations in almost every European country, but the EU directive warrants, that “*women have the right to a minimum of 14 weeks of maternity leave, of which at least two weeks are compulsory, and can be allocated before and/or after giving birth*” [92]. Recently, many European countries have made changes to the design of maternity leave provisions in order to allow for shared responsibilities for care of the newborn between both parents [92]. Nevertheless, the core of these regulations requires the health of pregnant and breastfeeding women to be sheltered, allow for a secured income and defend against dismissal. Precise conditions concerning work hours and kind of work (avoiding toxics or heavy lifting, e.g.) are listed so that an evaluation of danger can be performed for the pregnant worker. The law in German-speaking countries differs between work that cannot be carried out because of pathology or during pregnancy (which then induces work inability to be paid for by the health insurance) and inappropriate workplace conditions (which induce a ban from work until the employer moves the woman to a suitable workplace) [92]. A cooperative action is therefore needed, involving both a gynecologist as well as an occupational health specialist to find a solution. In Lausanne (Switzerland), a clearing office was installed in 2015, which offers consultation and contacts employers in those cases [93]. This example shows that if the law is followed in a careful and acceptable way and women see that their interests are cared for, they do return to work [93]. However, there are still a lot of problems to be solved with the translation of maternity leave policies into reality, even in Europe. Just to name one example, all these benefits only apply to insured persons, i.e., those working in precarious work conditions cannot benefit from them [93]. 

It is becoming increasingly clear that society can no longer leave the consequences of childbearing to women alone since these well-trained and graduated mothers are needed to run the economy. Breastfeeding and the development of breastfeeding-promoting environments must therefore be perceived as a task for society as a whole, which includes the identification and removal of barriers to breastfeeding [19,20].

### 4.2. Strengths and Limitations

This present study revealed a number of strengths and limitations. The strengths can be categorized into three dimensions: -This present paper includes a review from a German point of view, integrating aspects of social law into a framing model. It is the first paper on this topic that the authors are aware of.-The publication complements the state of research in so far as it combines gynecological, public health and teaching-related implications. These arise from the expertise of the three authors. S.H.B. is a trained gynecologist with over 30 years of experience in social medicine. H.A. has many years of practical experience in the field of obstetrics at a level 1 perinatal center, where he works as its medical director. He also heads a teaching and research unit in the field of midwifery. J.G. is an expert in public health, with a focus on women’s health. He also brings in aspects of professional education based on years of teaching experience in the master course of midwifery at Tübingen University.-The paper provides a current summary of the existing state of research, including broad access to the relevant literature with identifying more aspects that Patil et al. [22], especially regarding not only barriers but also enablers of breastfeeding.

There are also some limitations in this study that could distort the results. Firstly, it remains unclear whether the inclusion and exclusion criteria and the search strategy were sufficiently valid to identify all relevant studies. Since the search was only conducted using PubMed, articles in non-PubMed-listed journals could not be identified. Only English-language articles were included. A further limitation is that the quality of the *n* = 57 included studies was not evaluated using validated assessment tools. However, the critical review of the review by Carpay et al. (2021) [27] underlines the need to always analyze the methodological and content quality. This present work is not a systematic review, for which there are correspondingly stricter requirements. Limitations also arise from the design of this present study, which was conceived as a narrative review. Narrative reviews have inherent limitations in terms of objectivity, completeness of literature search and interpretation of findings [94]. It should be noted that this present study also aimed to validate and extend the model developed by Patil et al. (2020) [22], which indicates that the limitations of this study [22] also had a limiting effect in the present study. It should also be noted that Patil et al. (2020) [22] included studies that looked at exclusive breastfeeding, whereas this study examined the barriers and enablers of breastfeeding in general.

## 5. Conclusions

It can be stated that there are many barriers that explain why the majority of newborns are not breastfed according to WHO recommendations [12,13,14,15,16]. However, as shown in Table 2, there are also many enablers that need to be expanded within all levels of society in the interests of both children’s and women’s health. Breastfeeding promotion must be perceived as a task for society as a whole [19,20], which implies that specific, low-threshold services must also be made available for all groups at particular risk (e.g., mothers of preterm births or twins). Breastfeeding has to be learned by mother and child and breastfeeding problems often occur at the beginning, which makes it difficult for young mothers to persevere; therefore, patient, sensitive guidance from experienced midwives is important during the first few weeks post-partum. This intervention is also offered as part of midwifery care—but the severe shortage of midwives in many places does not allow for sufficient care to be provided for all first-time mothers. In addition, this professional group not only has to fulfill a variety of obligations in the context of midwifery care, freelance midwives also run birth centers and attend home births [95]. Nevertheless, the diverse and sometimes complex interrelationships of existing breastfeeding barriers must be taught at an early stage during midwifery studies [96,97]. Health professionals who have completed their training must also receive regular training in woman-centered breastfeeding support, as some sources have shown that these professional groups also have significant knowledge gaps [61,63,64,65].

Better knowledge about the detrimental effects of formula feeding can also trickle into minds over time and help to abolish discharge packs containing formula. Taking this into account, the Neo-Milk project began in Germany on 1 January 2021. This scientific initiative is funded by the Innovation Fund of the Federal Joint Committee and aims to establish a structured breastfeeding promotion program and donor milk banks in neonatal intensive care units. The long-term goal is to ensure that every premature baby weighing less than 1500 g has access to mother’s milk or donor milk from the first day of life [98]. The authors consider that it was worth spending many months studying the literature: the new aspects increase hope for further helpful developments and initiatives to increase both breastfeeding initiation and continuation.

## Figures and Tables

**Figure 1 healthcare-12-02418-f001:**
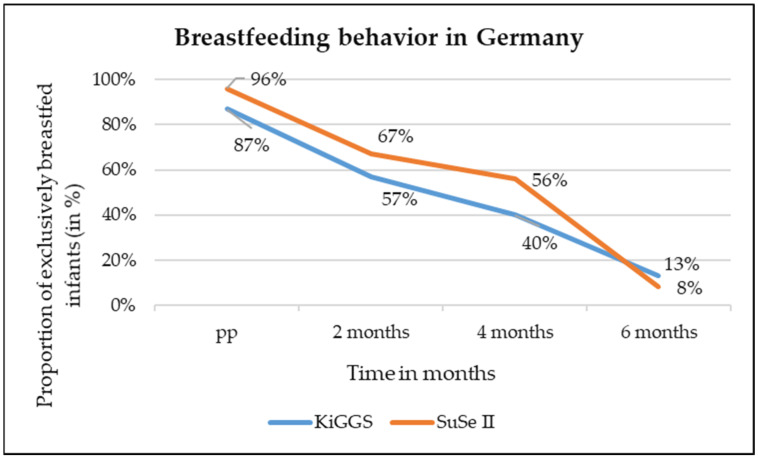
Changes in breastfeeding rates from birth to 6 months of age in Germany.

**Figure 2 healthcare-12-02418-f002:**
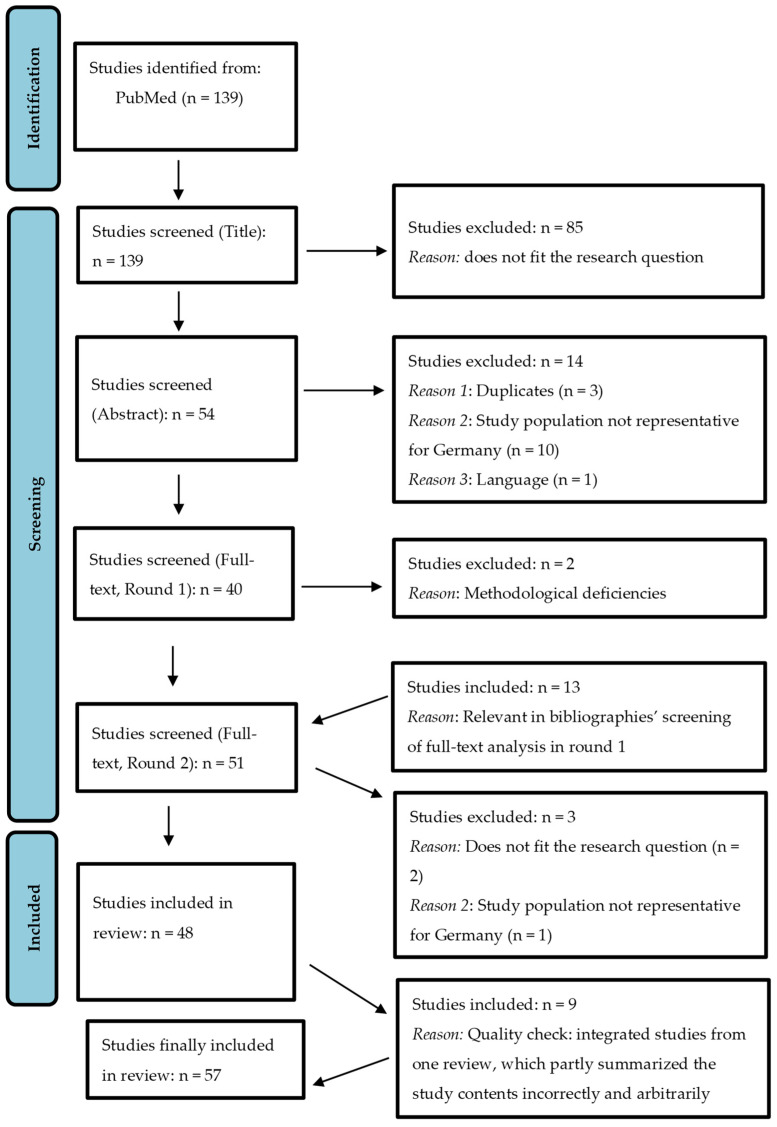
PRISMA flowchart of the literature synthesis (based on Ref. [26]).

**Table 1 healthcare-12-02418-t001:** List of barriers to breastfeeding from the literature.

Barriers: Main Categories			Barriers: Specifications	Type of Study	Source
**Individual-level factors**	**Attributes of the mother**				
		Maternal health	Psychological instability	Qualitative	Hinsliff-Smith et al. (2014), UK from Patil et al. (2020) [22]
			Medically complex pregnancies	Quantitative	Kozhimannil et al. (2014), USA [37]
			** Maternal obesity **	** Qualitative **	** Keely et al. (2015), UK [38] **
				** Quantitative **	** Kozhimannil et al. (2014), USA [37] **
				** Quantitative **	** Kair and Colaizy (2016a), USA [39] **
			Use of formula during the first month	Quantitative	Mallan et al. (2018), Australia [40]
		Changes in figure and breast shape	Fear of damaging appearance	Quantitative	Shepherd et al. (2017), UK from Patil et al. (2020) [22]
		**Insufficient breast milk**		**Mixed-methods study**	**Teich et al. (2014), USA** **from Patil et al. (2020) [22]**
				**Quantitative**	**Shepherd et al. (2017), UK from Patil et al. (2020) [22]**
				** Quantitative **	** Kair and Colaizy (2016a), USA [39] **
		Cesarean section birth	Infants were formula-fed while mother recovered from surgery, no initiation of breastfeeding within the first hour after birth	Mixed-methods study	Teich et al. (2014), USA from Patil et al. (2020) [22]
		**Lack of mothers’ knowledge on breastfeeding practices and benefits**		**Qualitative**	**Nesbitt et al. (2012), Canada** **from Patil et al. (2020) [22]**
				**Qualitative**	**Hinsliff-Smith et al. (2014), UK** **from Patil et al. (2020) [22]**
			No latching skills	**Quantitative**	**Taveras et al. (2004), USA** **from Patil et al. (2020) [22]**
		**Smoking during pregnancy**	OR in meta-analysis (together with one more study from Chile): 2.49 (95% CI 2.16–2.89)	**Quantitative**	**Blomquist et al. (1994), Sweden** **from Patil et al. (2020) [22]**
				**Quantitative**	**Ludvigsson and Ludvigsson (2005), Sweden** **from Patil et al. (2020) [22]**
				**Quantitative**	**Kristiansen et al. (2010), Norway, from Patil et al. (2020) [22]**
				**Quantitative**	** Demirci et al. (2013), USA [44] **
	**Attributes of the infant**	Initial weight loss		Quantitative	Blomquist et al. (1994), Sweden from Patil et al. (2020) [22]
		Medically complex children		Mixed-methods study	Hookway et al. (2021), UK [51]
		** Preterm birth **	** Maternal frustration with separation/breastfeeding difficulties **	** Quantitative **	** Kair and Colaizy (2016b), USA [47] **
				** Quantitative **	** Gianni et al. (2018), Italy [50] **
				** Qualitative **	** Yang et al. (2019), China [48] **
			** Longer stay in hospital with resulting separation from the mother and/or formula use on the ward **	** Quantitative **	** Mattsson et al. (2015), Sweden [46] **
				** Quantitative **	** Gianni et al. (2016), Italy [49] **
				** Quantitative **	** Jónsdóttir et al. (2020), Iceland [28] **
**Group-level factors**	Hospital and health services	**Inappropriate communication from healthcare staff**		**Qualitative**	**Hinsliff-Smith et al. (2014), UK** **from Patil et al. (2020) [22]**
				**Mixed-methods study**	**Teich et al. (2014), USA** **from Patil et al. (2020) [22]**
			Negative advice related to EBF	**Quantitative**	**Taveras et al. (2004), USA** **from Patil et al. (2020) [22]**
		** Lack of knowledge/ ** ** educational need for staff **		Qualitative	Desmond and Meaney (2016), Ireland [70]
				Qualitative	Michaud-Létournaud et al. (2022), Canada [64]
			Poor culture of supporting breastfeeding	Quantitative	Hookway and Brown (2023), UK [63]
	**Home/family environment**	Impact of social and intimate relationships	Teenage mothers	Qualitative	Nesbitt et al. (2012), Canada from Patil et al. (2020) [22]
		negative influence from family members and society		Qualitative	Nesbitt et al. (2012), Canada from Patil et al. (2020) [22]
			Family-run enterprise does not respect maternity leave,	Qualitative	González-Pascual et al. (2017), Spain [83]
	**Work environment**	**Resuming work/school**	return to work warrants an established feeding routine	Qualitative	Desmond and Meaney (2016), Ireland [70]
				Quantitative	Ludvigsson and Ludvigsson (2005), Sweden from Patil et al. (2020) [22]
			** Lack of support from colleagues, rigid schedules, inappropriate location to pump **	** Quantitative **	** Sattari et al. (2013), USA [71] **
				** Quantitative **	** Hendrickson et al. (2022), USA [77] **
				** Quantitative **	** Nourse (2024), USA [79] **
		** Unpaid maternity leave **	** realization left up to the individual employer’s approach, too many exemptions **	** Qualitative **	** ACOG (2021), USA [88] **
				** Qualitative **	** Rykiel et al. (2023), USA [85] **
				** Qualitative **	** Taylor (2023), UK [82] **
	Community environment	Lack of privacy	At home and in public spaces	Qualitative	Nesbitt et al. (2012), Canada from Patil et al. (2020) [22]
				Qualitative	Hinsliff-Smith et al. (2014), UK from Patil et al. (2020) [22]
**Society-level factors**	Parents‘ ages	Maternal age < 25		Quantitative	Blomquist et al. (1994), Sweden from Patil et al. (2020) [22]
		Father’s age < 30		Quantitative	Ludvigsson and Ludvigsson, (2005), Sweden from Patil et al. (2020) [22]
	Parents‘ education			Quantitative	Ludvigsson and Ludvigsson, (2005), Sweden from Patil et al. (2020) [22]
				Quantitative	Demirci et al. (2013), USA [44]
	**Single marital status/** **single mothers**			**Quantitative**	**Dennis et al., 2014 Canada** **from Patil et al. (2020) [22]**
				**Quantitative**	** Demirci et al. (2013), USA [44] **
				**Quantitative**	** Jónsdóttir et al. (2020), Iceland [28] **
	Negative influence of mass media	As a source of information		Quantitative	Pechlivani et al. (2005), Greece from Patil et al. (2020) [22]
	** Maternal discharge packs containing formula **			** Qualitative **	**Dunn et al. (2015a), USA [29]**
				** Qualitative **	** Sriraman et al. (2016), USA [86] **
				** Qualitative **	** ACOG (2021), USA [88] **
	Policy/legal regulation	Essential failure to meet legal standard	Shortcomings with existing law: Affordable Care Act	Qualitative	Majee et al. (2016), USA [69]
				Qualitative	ACOG (2021), USA [88]

ACOG = American College of Obstetricians and Gynecologists; CI = confidence interval; EBF = exclusive breast feeding; OR = odds ratio. Bold characters: certainty of statements [31]; aspects highlighted in blue: extension of the state of research presented by Patil et al. (2020) [22].

**Table 2 healthcare-12-02418-t002:** List of enablers to breastfeeding from the literature.

Enablers: Main Categories			Enablers: Specifications	Type of Study	Source
**Individual-level factors**	**Attributes of the mother**	Self-efficacy, pride and regret		Quantitative	Shepherd et al. (2017), UK from Patil et al. (2020) [22]
				Quantitative	Gerhardsson et al. (2018), Sweden [52]
		All forms of timely and skilled support		Quantitative	McFadden et al. (2017), UK [32]
		knowledge, attitude, subjective norm and practice control		Quantitative	Zhang et al. (2018), China [33]
		Educational initiative	Significantly increased knowledge of breastfeeding after training	Quantitative	Iliadou et al. (2018), Greece [34]
	**Attributes to the infant**	Preterm birth in hospital care	** Breastfeeding support by healthcare professionals **	** Quantitative **	** Goyal et al. (2014), USA [43] **
				** Quantitative **	** Gianni et al. (2018), Italy [50] **
				** Quantitative **	** Mitha et al. (2019), France [55] **
				** Quantitative **	** Estalella et al. (2020), Spain [60] **
			Kangoroo Care/skin-to-skin contact	RCT (does not reach significance due to power problem)	Hake-Brooks and Anderson (2008), USA [53]
					Mörelius et al. (2015), Sweden [54]
				Quantitative	Crippa et al. (2019), Italy [45]
				Quantitative	Mitha et al. (2019), France [55]
		Preterm birth after discharge	Peer support by social media did not work in groups	RCT	Niela-Vilén et al. (2016), Finland [56]
			Telephone support after discharge reduces parental stress	RCT	Ericson et al. (2018), Sweden [57]
				RCT	Ericson et al. (2019), Sweden [58]
			Support by web-based education program and home visits after discharge helps	Quantitative	Jang and Hong (2020), Korea [59]
			Online-education and support within 7 days pp increases knowledge	Quantitative	Zhang et al. (2024), China [61]
**Group level factors**	**Hospital and health services**	Rooming-in and feeding on demand		Quantitative	Pechlivani et al. (2005), Greece from Patil et al. (2020) [22]
		**Healthcare staff**	Support from the nurses in early postpartum period	Qualitative	Hinsliff-Smith et al. (2014), UK from Patil et al. (2020) [22]
				Qualitative	Nesbitt et al. (2012), Canada from Patil et al. (2020) [22]
			Support in hospital, at home or in community	Quantitative	Rayfield et al. (2015), UK [42]
		Educational initiative	significantly increased knowledge of breastfeeding after training	Quantitative	Ramirez-Duran et al. (2024), Spain [65]
	**Work environment**	Resuming work/school	90% who breastfed did so for 6 months or longer (very supportive environment)	Quantitative	Melnitchouk et al. (2018) USA [75]
			EBF rate for 6 months: 63.6%	Quantitative	Nanthakomon et al. (2023), Thailand [81]
		Breastfeeding-specific policy in the workplace,	Treats breastfeeding as a category of applied occupational medicine	Qualitative	Taylor (2023), UK [82]
	**Community environment**	seamless continuity of care between hospital and home	Establish peer counselor contact before discharge for late preterms	Qualitative	Bennett and Grassley (2017), USA [84]

RCT = randomized controlled trial. Bold characters: certainty of statements [31]; aspects highlighted in blue: extension of the state of research presented by Patil et al. (2020) [22].

## Data Availability

The raw data supporting the conclusions of this article will be made available by the authors on request.

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
