# Peer review of "Challenges and Choices in Breastfeeding Healthy, Sick and Preterm Babies: Review"

_healthcare, 2024, doi:10.3390/healthcare12232418_

Round 1

Reviewer 1 Report

Comments and Suggestions for Authors

The topic is very interested, However the way it is presented is very confusing. There is no clear aim and output of the study design, even though it is a narrative review.

Introduction: Data presented is very limited to this section. The author could present cumulative various data and barriers issues from Europe, Asia, Africa, America and not only selection from 2-3 countries. Data should include, prevalence of breasting feeding and exclusive breastfeeding in those continents, different Barries (old and new ones).

Aim of study: the aim of study is not clear at all. The research question is also very confusing. Author should restate the research question and the purpose of the study in a simple clear manner. (2-3 sentences max). If the authors wants to address 2-3 research questions this is fine with a narrative review but make sure those questions are clear.

The PRISMA method is very well known to be used in systematic reviews and not in Narrative. What is the purpose of this study to use PRISMA? I do not see any benefit, rather I would say it confused me more to understand the focus of the review.

Literature search: The narrative reviews do not typically involve  inclusion and exclusion criteria. What is the reason and benefit the author have chosen these? 

Authors should outline how conducted analyses and how they determined that sufficient analyses and interpretation was achieved. 

The authors have stated the liminations of the review. What about the strengths of the study?

In general the review paper is extremely long with not clear focus. I would suggest to limit the total number of pages to 25 max.

Reviewer 2 Report

Comments and Suggestions for Authors

Dear Authors,

In the manuscript entitled “Challenges and choices in breastfeeding in healthy, sick and preterm babies: Review” the authors have tried to study the effects of breastfeeding in healthy, sick and preterm babies. My overall evaluation is negative. There are a number of major revision, formal and scientific aspects that should be addressed.  

1.      Figure 1 was not available for review. It is necessary to check if it has been uploaded.

2.      According to the type of writing of the article, it seems that it is not a narrative review.  It is necessary for the authors to provide more explanations to the reader about how to write in the text of the article.

3.      Why is the article not organized according to the systematic review with the aim of conceptual model.

4.      Regarding the topic of the article, the authors do not clearly specify how to solve the problem. For example, in part 5. Conclusions, they talk about the milk bank, but in the text, they do not mention its importance and how to use it. Therefore, it is necessary to discuss the specifications, receiving and preparation of donated milk in the text.

5.      According to the title of the article, it is necessary to add information about the effects of stress on breast milk and how to manage it, especially in mothers with premature babies.

6.      It is necessary to add a question about the services that the hospital can provide to these mothers in the text of the article.

7.      The article talks about reducing the risk of cancer in breastfed babies, but if the mother has a metabolic disease such as obesity, diabetes or cancer, what measures should be taken for their children is not discussed. There is no discussion about the quality of breast milk, how it should be evaluated to meet the needs of the child, and it is necessary to add some information in this regard in the article.

Round 2

Reviewer 1 Report

Comments and Suggestions for Authors

The authors have successfully responded to all my comments

Reviewer 2 Report

Comments and Suggestions for Authors

Dear Authors

According to the authors' explanations, I understood their purpose in writing this article. However, it would have been better if they had completed the current text, considering the issues I raised. However, the overall article is acceptable in my opinion. In addition, Figure 1 is still not visible in the current version.